# Achiral hard bananas assemble double-twist skyrmions and blue phases

Rodolfo Subert [1], Gerardo Campos-Villalobos [1] & Marjolein Dijkstra [1,2] ✉

Skyrmions are topologically protected, vortex-like structures found in various condensed-matter systems including helical ferromagnets and liquid crystals, typically arising from chiral interactions. Using extensive particle-based simulations, we demonstrate that non-chiral hard banana-shaped particles, governed solely by excluded-volume interactions, spontaneously stabilize skyrmion structures through the bend-flexoelectric effect. Under thin confinement, we observe the formation of quasi-2D layers of isolated skyrmions or dense skyrmion lattices. These structures, comprising a racemic mixture of left- and right-handed skyrmions, show resilience against thermal fluctuations while remaining responsive to external fields, offering intriguing possibilities for manipulation. We also find that the size of these skyrmions can be adjusted by the dimensions and curvature of the banana-shaped particles. In the absence of geometric frustration due to confinement, a blue phase III may emerge, characterized by a 3D network of chiral skyrmion filaments of the nematic director field within an isotropic background. Our findings provide valuable insights into stabilizing skyrmion lattices and blue phases, showcasing non-Gaussian fluid-like dynamics in systems of achiral hard particles. Furthermore, they highlight the remarkable capacity of these complex fluids in designing advanced functional materials with diverse applications in photonics and memory devices.

Skyrmions were introduced by Tony Skyrme in 1961 to describe the nucleus through convoluted twists within the quantum field generated by quark-antiquark pairs, also known as pions[1]. Originally developed for nuclear physics, the skyrmion model has extended its relevance to elucidate similar complex structures observed in condensed-matter systems like quantum Hall magnets and Bose-Einstein condensates[2,3]. Notably, within helical ferromagnets, skyrmions adopt a chiral spin structure with a swirling configuration. These structures are topologically protected as a (free) energy barrier has to be overcome to remove the vortex-like structure[4]. Hence, these systems have potential applications for information storage and processing. The emergence of skyrmions in such systems is attributed to the absence of inversion symmetry (chirality) and the presence of Dzyaloshinskii-Moriya spin-orbit interactions[5].

More recently, there has been significant interest in skyrmions originating in another class of materials governed by chiral interactions, namely highly chiral liquid crystals, which predominantly yield chiral nematic or so-called cholesteric (N*) phases. These phases lack positional order, but exhibit orientational order as particles align preferentially along a so-called nematic director. In the case of chiral nematics, the nematic director $\hat{\boldsymbol{n}}(\mathbf{r})$ rotates as a helix around a single chiral director, resulting in a chiral distribution of particle orientations with a cholesteric pitch $\mathcal{P}$ representing the length scale associated with the helical periodicity.

[1]Soft Condensed Matter & Biophysics, Debye Institute for Nanomaterials Science, Utrecht University, Princetonplein 1, 3584 CC Utrecht, The Netherlands. [2]International Institute for Sustainability with Knotted Chiral Meta Matter (WPI-SKCM2), Hiroshima University, 1-3-1 Kagamiyama, 739-8526 Higashi-Hiroshima, Hiroshima, Japan. ✉e-mail: m.dijkstra@uu.nl

In these systems, skyrmion structures were observed over a century ago[6], albeit indirectly, in the form of filaments within the so-called cholesteric blue phases (BPs)[7–10]. These phases, typically formed by highly chiral liquid crystals, exhibit a strong local preference for a double-twist structure over the single-twist arrangement seen in a helical cholesteric phase. To illustrate this, envision a single mesogen aligned along the axis of a cylinder. Other mesogens can surround it radially with a slightly tilted particle orientation[11]. This radial arrangement gives rise to a double-twisted tubular structure known as a double-twist cylinder (DTC) or skyrmion filament. The curved line segments in Fig. 1c–d represent the nematic director field profile of a skyrmion filament core. While DTCs are the local ground state for chiral particles[12], they cannot fill the entire space, and topological defects or disclination lines are necessary to stabilize these structures[5]. Consequently, BPs can only exist at sufficiently high twisting power, where the gain in free energy due to the formation of DTCs counterbalances the loss in free energy caused by the formation of defects in $\hat{n}(\mathbf{r})$. The loss in free energy due to defects is minimal near the isotropic fluid phase, explaining the appearance of BPs close to the isotropic-cholesteric phase boundary.

Three types of BPs consisting of networks of DTCs and disclinations have been observed experimentally[10,13–16]. BPI and BPII consist of rigid DTCs arranged in body-centered cubic and simple cubic lattices, respectively, intertwined with a lattice of topological defect lines. The structure of the third blue phase, BPIII, posed more difficulties and was only recently unveiled[17], establishing it as a liquid tangle of DTCs or skyrmion filaments. This finding explains its hazy appearance, which has led to its alternative name "blue fog". Direct observation of DTCs within bulk BPIII is challenging due to the small size of the cylinders and the strong thermal fluctuations in this phase. Recent experimental studies have investigated the skyrmion structure of BPs by confining the materials into very thin layers[10,17]. The cholesteric pitch in these confined systems cannot be completely relaxed, causing the DTCs to align their axis perpendicular to the confining walls. Due to this alignment, the molecules can form either a quasi-2D fluid of 2D skyrmions or a hexagonal skyrmion lattice (SL)[17,18]. The 2D skyrmions are often referred to as "baby skyrmions" because of their low-dimensional topological similarity to higher-dimensional Skyrme solitons[10].

The emergence of helical phases is not exclusive to cholesteric liquid crystals. A remarkable, experimentally discovered liquid-crystal phase with a new type of order and exhibiting a helical structure is the twist-bend nematic (N$_{TB}$) phase formed by bent-core mesogens[19–22]. Recently, this phase has also been observed in simulations of colloidal (polar) banana-shaped particles[23–25]. In the N$_{TB}$ phase, the nematic director field rotates around a right circular cone, as predicted earlier by Meyer[26] and Dozov[27], and hence the N$_{TB}$ phase is chiral even though the particles are achiral. The emergence of spatial modulations in the nematic director field of the N$_{TB}$ phase is attributed to the bend-flexoelectric effect, in which spontaneous bend deformations induce a net polarization. The polar order may be the electrostatic, magnetic, or steric ordering of molecular or particle shape[18]. To resolve the problem that pure bend deformations cannot uniformly fill three-dimensional space, local bend deformations must be accompanied by either a spontaneous twist, resulting in an N$_{TB}$ phase, or splay distortions, leading to a splay-bend nematic (N$_{SB}$) phase. The latter phase has recently been observed in experiments with colloidal bananas and bent silica rods[28,29].

Considering the remarkable similarities between the modulated N* and N$_{TB}$ phases, a fundamental question emerges: Can skyrmions exist in systems composed of achiral curved particles? Recent research efforts have addressed the general question of whether bend-flexoelectric coupling could potentially stabilize complex modulated liquid crystal phases, like BPs, using varying theoretical frameworks[18,30,31]. These theoretical investigations suggest that complex skyrmion-like structures minimize the free energy within a narrow regime in the phase diagram[18] or exhibit some degree of metastability[31]. Interestingly, a recent experimental study has shown that colloidal bananas can form a two-dimensional vortex phase, which notably differs from true skyrmions and is driven by the polydispersity

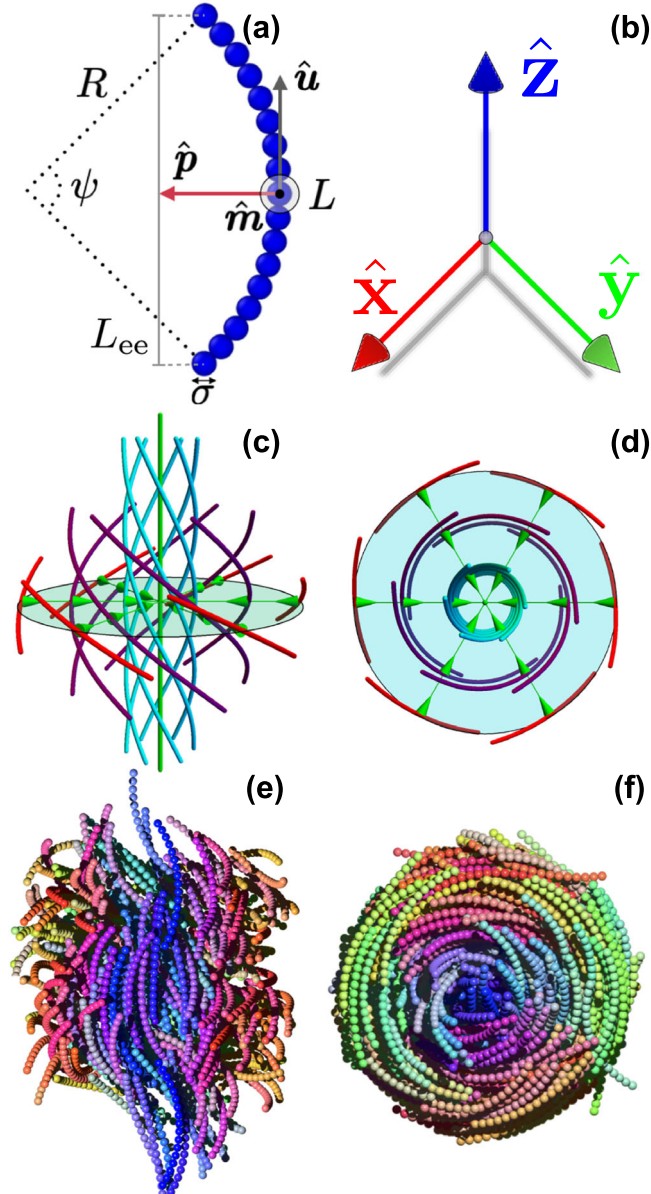

**Fig. 1 | Schematics of skyrmion filaments of achiral banana-shaped particles. a** Hard curved spherocylinder with a length-to-diameter ratio $L/\sigma = 16$ modeled by tangent hard spherical beads of diameter $\sigma$ and opening angle $\psi = L/R = 1.6$ with $R$ the radius of curvature. The long $\hat{u}$ and polar $\hat{p}$ axes of the particle are represented by the gray and red arrows. The third particle axis is simply $\hat{m} = \hat{u} \times \hat{p}$. **b** Color legend for banana-shaped particles. The particles in the displayed configurations are colored according to the orientation of their long $\hat{u}$ particle axis using the RGB color coding as shown in the legend. **c, d** Nematic director field lines $\hat{n}(\mathbf{r})$ of a skyrmion filament or double twist cylinder seen in perspective and from above, respectively. The orientation of the long axis $\hat{u}$ of individual particles is expected to align with the field lines, vertical and continuous at the core and tilted for higher radii. In the case of polar particles, the skyrmion filament also exhibits a polarization field $\mathbf{P}(\mathbf{r})$ (green arrows), which is not defined at the core of the skyrmion filament, identifying a $\lambda$-line in the center. This polarization field consistently aligns with the direction of the bend vector of the field lines. Therefore, we can identify the core as a discontinuity in the bend field, known as $\beta$-line[31]. **e, f** Typical configuration of a skyrmion filament of banana-shaped particles obtained from our simulations, displaying both cut-through and top perspectives. The simulations remarkably align with the theoretical predictions.

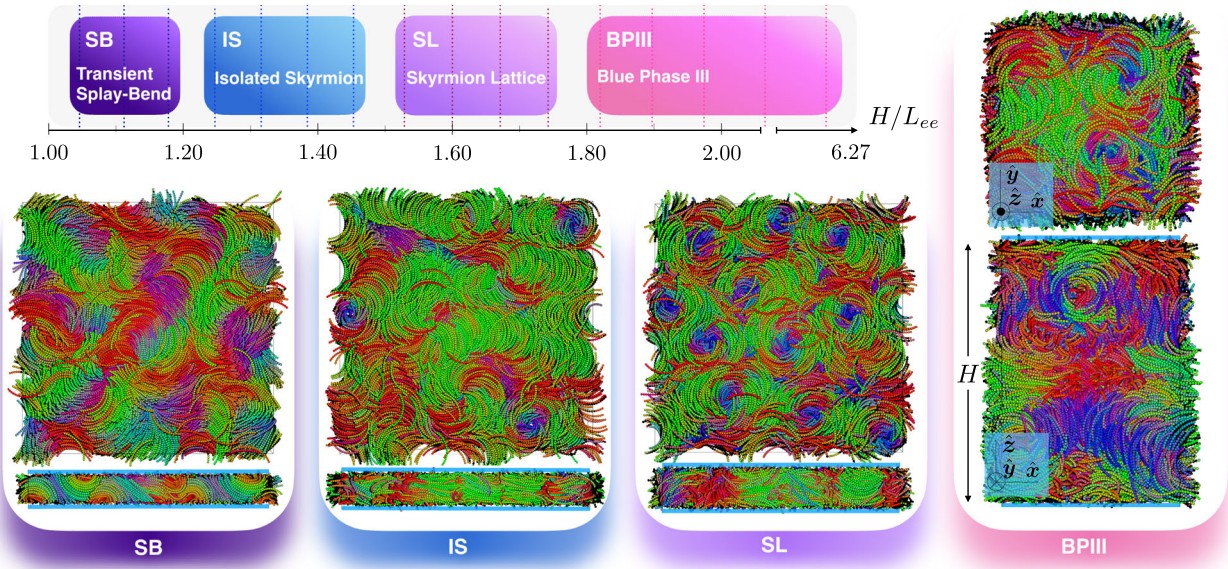

**Fig. 2 | Phase sequence of banana-like particles confined in thin layers.** The equilibrium phases of banana-like particles with length-to-diameter ratio $L/\sigma = 16$ and opening angle $\psi = 1.6$ under one-dimensional confinement are shown as a function of the confinement height $H$ at a packing fraction $\eta \simeq 0.33$. Note that the values of $H$ are relative to the end-to-end length $L_{ee}$ of the particle. Top and side views of typical configurations representative of each phase are shown in the individual panels. The color of the particles is assigned according to their orientation in space. In the side views, the approximate position of the boundary walls is represented with the bold blue lines perpendicular to the $\hat{z}$ axis. Note that the systems are periodic in the $\hat{x}$ and $\hat{y}$ directions.

of the bananas, i.e., particles with high curvature tend to be closer to the center of the vortex[32]. While this observation highlights the potential of banana particles to spontaneously stabilize vortex-like textures, the existence of bend-stabilized skyrmions remains unconfirmed, lacking both experimental validation and support through particle-based simulations.

In this Letter, we demonstrate the spontaneous formation of bend-stabilized skyrmions and corresponding blue phases in systems of achiral hard banana-shaped particles using extensive particle-based simulations. We demonstrate how entropy and the bend-flexoelectric effect induced by the unique shape of achiral colloidal banana-shaped particles stabilize fractional skyrmion filaments, which exhibit chiral symmetry breaking. Similar to the behavior observed in the $N_{TB}$ phase, we show that these bent particles orient their polar axes $\hat{p}$ towards the central axis of the skyrmion filament, contributing significantly to the stabilization of this distinctive structure. Consequently, the resulting skyrmion configuration within these systems is characterized by a polarization vector field that aligns with the bend vector of the nematic director integral curves, as depicted by the green arrows in Fig. 1c, d. Under thin confinement, we find two distinct structures a hexagonal lattice of skyrmions and a phase comprised of isolated skyrmions. In bulk, the skyrmion filaments become flexible, creating an intricate network, reminiscent of a BPIII phase before transitioning to the $N_{TB}$ phase when compressed from a low-density isotropic (I) phase.

## Results

### Bend-stabilized skyrmions in confinement

Inspired by experimental observations on the emergence of baby skyrmions in thin layers of cholesteric phases[10,17], we start our investigation by confining twist-bend nematic ($N_{TB}$) phases of banana-shaped particles to lateral dimensions smaller than their respective equilibrium pitch $\mathcal{P}_{TB}$, thereby frustrating the relaxation of the pitch. Drawing an analogy with cholesteric phases, we expect the spontaneous self-assembly of rigid skyrmion filaments perpendicular to the confining walls. To this end, we perform Molecular Dynamics (MD) simulations at constant temperature $T$, in which $N$ banana-shaped particles confined between two parallel soft walls are slowly

compressed laterally up to a packing fraction $\eta \approx 0.33$ at which, as we shall see below, the bulk $N_{TB}$ phase is stable. Each particle is modeled as a rigid body comprising 17 purely repulsive tangent beads with a diameter of $\sigma$. These beads are arranged on an arc segment of length $L = 16\sigma$ and a radius of curvature $R$, resulting in an opening angle of $\psi = L/R$ (see "Methods"). The individual beads interact via a pseudo hard-sphere potential, which accurately reproduces the thermodynamic and structural properties of hard-sphere fluids[33]. The particle orientation is described by three unit vectors, $\hat{u}$, $\hat{p}$, and $\hat{m}$, indicating the orientations of its long, medium, and short axes, as depicted in Fig. 1a.

Given that previous studies have established that chiral blue phases appear as intermediate phases between the I and the (helical) N* phase, we select banana-like particles with an opening angle $\psi = 1.6$ and an end-to-end length $L_{ee} = 2R\sin(\psi/2) \sim 14.4\sigma$. It has been demonstrated that within the range $\psi \in [1.2, 1.9]$ these particles exhibit a direct transition from the I to the $N_{TB}$ phase[25,34], while for lower values of the opening angle, $\psi < 1.2$, the particles show an intermediate uniaxial nematic phase between the I and $N_{TB}$ phase.

We study the phase sequence as a function of the confinement height $H$, and present the results in Fig. 2. We observe in order of increasing thickness $H$, a splay-bend-like (SB) phase, isolated skyrmions (IS), a quasi-2D lattice of hexagonally packed skyrmions (SL), and a BPIII-like phase where flexible DTCs are confined between the soft walls. The phase sequence is remarkably similar to that reported in ref. 10 for an experimental system of cholesteric liquid crystals, and this is due to an interplay of confinement and boundary conditions. Specifically, when the confinement height $H$ is smaller than the end-to-end length $L_{ee}$ of the banana-shaped particles, i.e., $H \lesssim L_{ee}$, the degenerate planar anchoring enforced by the walls leads to a splay-bend-like (SB) modulation. As the thickness increases to $L_{ee} < H < 2r_s$, with $r_s$ the radius of the skyrmion filaments, the banana-shaped particles nucleate DTCs that align perpendicularly to the walls. At sufficiently low confinement heights, only a few isolated skyrmions (IS) are randomly nucleated, forming a dilute, gas-like phase of skyrmions. The mobility of the vortices is determined by the available space to relax the DTC pitches, and at larger thicknesses, the DTCs pack into a quasi-2D lattice

of hexagonally packed skyrmions (SL). It is important to note that the skyrmion lattice in our system is a racemic mixture of left-handed and right-handed skyrmions. Supplementary Fig. 1 shows this mixture, with a slight asymmetry in handedness due to small system size fluctuations. The formation of a hexagonal skyrmion lattice (SL) results from the achirality of the skyrmion-skyrmion interaction: regardless of chirality, particles on the edge of a half-skyrmion align their long axes perpendicular to the skyrmion axis and point their polar axes inward. This makes the skyrmion interaction independent of chirality, allowing the formation of a hexagonal lattice. In contrast, for quarter-skyrmions (see ref. 35), where particles at the boundary lie at 45° with respect to the axis, chirality becomes crucial in skyrmion interactions, allowing only skyrmions with opposite chirality to neighbor each other. The resulting equilibrium lattice would then be an "antiferromagnetic" square lattice with a perfect balance between left- and right-handed skyrmions. Finally, as the thickness approaches the equilibrium pitch, another phase forms, where long and flexible DTCs are bounded by the confining walls. As it will be discussed next, such a phase is reminiscent of the BPIII[36], which to date, has only been found in thermotropic cholesteric liquid crystals.

## Structure of skyrmions

To investigate in more detail the structure of skyrmionic filaments and to acquire reliable statistical data, we perform additional simulations employing periodic boundary conditions along all three axes. This setup ensures the continuity of skyrmions across the entire cell and enables a clearer characterization and analysis of the structure of the skyrmionic filaments and the properties of the skyrmion lattices. To impose lateral confinement, we limit the thickness of the simulation box to $21\sigma$, resulting in a confinement height of $H/L_{ee} \sim 1.5$. The resulting configuration exhibits a well-defined hexagonal lattice of rigid skyrmions, as shown in Fig. 3a and Supplementary Movie 1 for the nucleation of a skyrmion lattice from the $I$ phase. Each skyrmion is accompanied by its corresponding pair of $\tau$-lines, resembling those observed by Pišljar and Nych[10,13].

To explore the emergent structures and identify DTCs, we expand upon previous techniques[10,37,38] that were able to successfully capture the gradual decrease in local alignment from the center of a DTC to its periphery. We compute both the local nematic order parameter $S$ and the local heliconical order parameter $\mathcal{T}$ for the local environment of each particle, which correspond to the largest (in absolute value) eigenvalues of the tensors $\mathbf{Q} = \langle \sum_i (3\hat{\mathbf{u}}_i \otimes \hat{\mathbf{u}}_i - \mathbb{I})/2N \rangle$ and $\mathbf{M}_2 = \langle \sum_i (\hat{\mathbf{u}}_i \otimes \hat{\mathbf{m}}_i + \hat{\mathbf{m}}_i \otimes \hat{\mathbf{u}}_i)/N \rangle$, respectively (see "Methods"). We define the local environment of a reference particle as all particles within a radial center-of-mass distance of $r_S = 10\sigma$ and $r_{\mathcal{T}} = 6\sigma$ for $S$ and $\mathcal{T}$, respectively. The use of these order parameters enables us to clearly identify the presence of orientational order within the system and identify and carve out the skyrmions without relying on external information regarding the skyrmion's orientation, or the system they belong to. By setting thresholds of $S < 0.18$ and $|\mathcal{T}| > 0.3$, we successfully identify the DTCs (see "Methods"). This methodology exclusively identifies bananas that belong to a skyrmion. In addition, the sign of $\mathcal{T}$ signifies the handedness of the skyrmions, allowing us to quantify the composition of the racemic mixture comprising both right- and left-handed skyrmions (see Supplementary Fig. 1).

We first present typical configurations of the axial and vertical cross-section of one of the skyrmion filaments in Fig. 1e, f. These configurations align remarkably well with the schematics in Fig. 1c, d. It clearly shows that the resulting structure is a half skyrmion as the director rotates by π/2 from the core to the boundary of the filament. In addition, due to the polarity of the particles, the skyrmion filament also exhibits a polarization field $\hat{\mathbf{P}}(\mathbf{r})$, which remains consistently perpendicular to the axis. This polarization field is undefined at the core of the skyrmion filament, corresponding to the predicted $\beta$-line at the center. This observation aligns well with both the $\beta$-lines in twist-bend

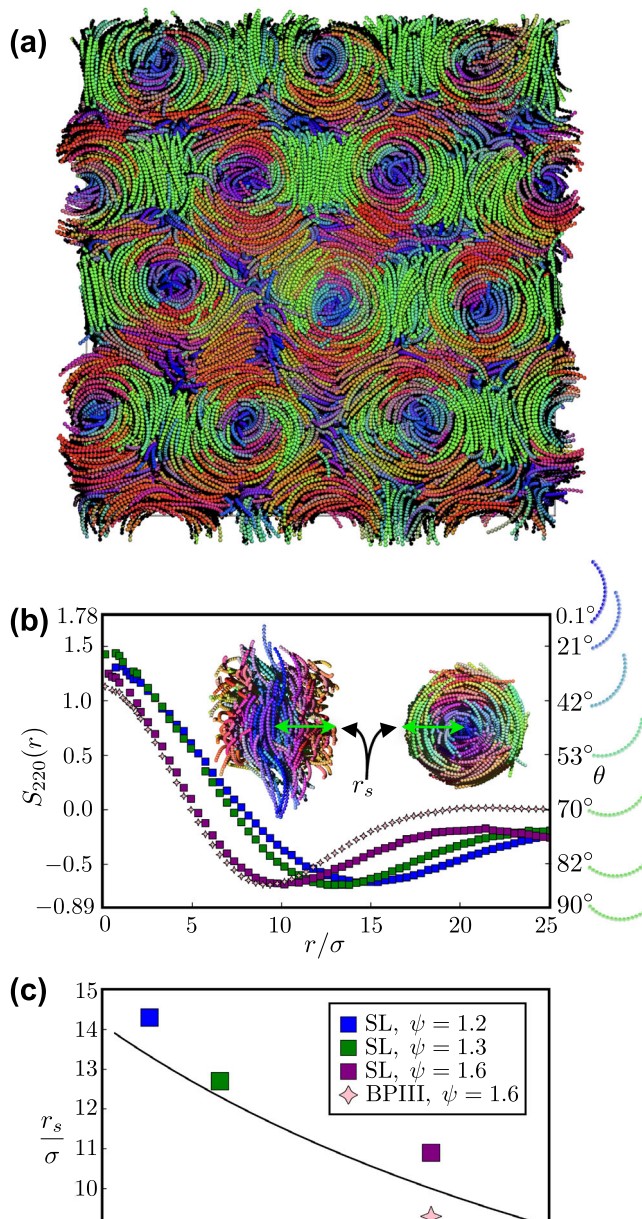

**Fig. 3 | Characterization of skyrmions of banana-shaped particles. a** Typical configuration of a lattice of hexagonally packed skyrmions (SL) as obtained from MD Simulations. **b** Orientation correlation function $S_{220}(r)$ of an SL phase consisting of banana-shaped particles with an opening angle $\psi = 1.2$, 1.3, and 1.6 and of a blue phase III (BPIII) with $\psi = 1.6$. The inset shows the top view and side view of a skyrmionic filament. The size of the skyrmion is determined by the radius $r_s$ as illustrated. The particles on the right-hand axis of the plot are colored according to a rotation around the $\hat{\mathbf{x}}$-axis. The top particle is parallel to $\hat{\mathbf{z}}$, the bottom one to $\hat{\mathbf{y}}$. **c** The radius of the skyrmion $r_s$ as a function of the opening angle $\psi$ of the banana-shaped particles for the SL and the BPIII phase. The solid line denotes the particle curvature $R/\sigma = L/\sigma\psi$ with $L$ the contour length and $\psi$ the opening angle of the banana. The standard error for each data point is about 1% or less and, as such omitted in the figure. Source data are provided as a Source Data file.

nematics as described by Alexander et al.[31], and the blue phase geometries identified by Shamid using lattice simulations[18]. Thus, our simulations clearly demonstrate that achiral bananas can spontaneously form bend-stabilized half skyrmions, also known as merons.

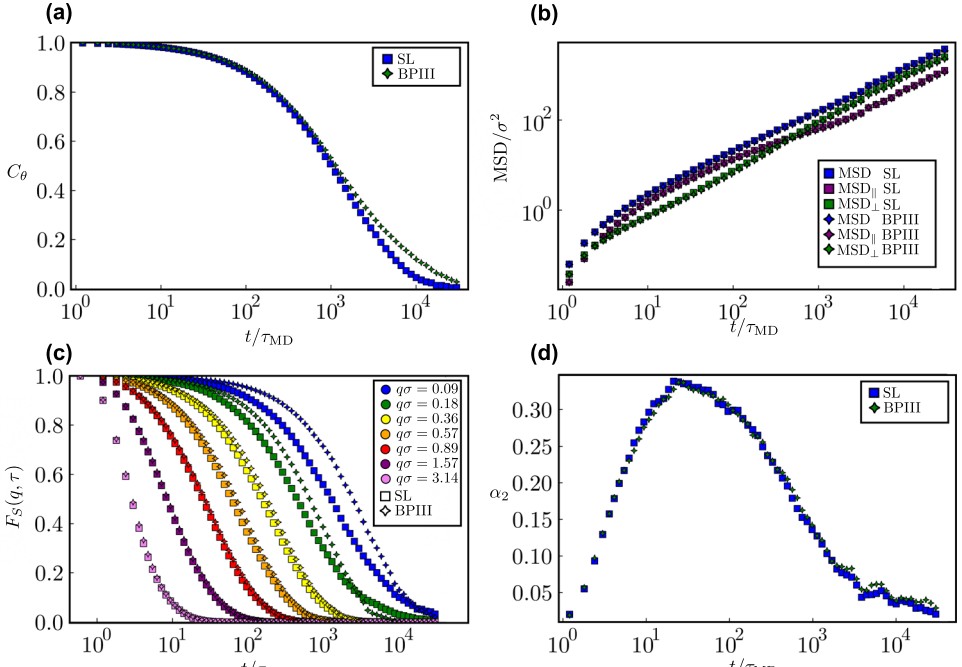

**Fig. 4 | Characterization of the dynamic behavior of a skyrmion lattice (SL) and blue phase III (BPIII) of banana-shaped particles. a** Orientational correlation function $C_\theta(t)$, (**b**) mean square displacements MSD($t$) along with its parallel MSD$_\parallel(t)$ and perpendicular MSD$_\perp(t)$ components, (**c**) intermediate scattering function $F_s(q, t)$ for varying wave vectors $q\sigma$ as labeled, and (**d**) non-Gaussian parameter $\alpha_2(t)$ as a function of time $t/\tau_{\text{MD}}$. Source data are provided as a Source Data file.

Moreover, achieving precise control over the size of skyrmions, characterized by their radius $r_s$, is crucial for potential applications. To calculate the average radius of skyrmion filaments, we use the orientation correlation function[39–41]

$$S_{220}(r) = \left\langle \sum_{i,\nu} \frac{S^*_{220}(\hat{\boldsymbol{u}}_i, \hat{\boldsymbol{a}}_\nu)\delta(r - r_{i\nu})}{S^*_{000}\delta(r - r_{i\nu})} \right\rangle, \qquad (1)$$

where $r_{i\nu}$ is the radial distance of a banana center-of-mass from the skyrmion axis and $\delta(r - r_{i\nu})$ is a delta function. Furthermore, $S^*_{000} = 1$ and hence, the denominator reduces to the standard radial distribution function, and $S^*_{220}(\hat{\boldsymbol{u}}_i, \hat{\boldsymbol{a}}_\nu) = \frac{2}{\sqrt{5}}(3(\hat{\boldsymbol{u}}_i \cdot \hat{\boldsymbol{a}}_\nu)^2 - 1)$ with $\hat{\boldsymbol{u}}_i$ the orientation of the long axis of particle $i$, and $\hat{\boldsymbol{a}}_\nu$ the orientation of skyrmion $\nu$ (see "Methods").

In Fig. 3b, we compare the orientation correlation functions $S_{220}(r)$ measured on hexagonally packed SLs of bananas with opening angles $\psi = 1.2$, 1.3, and 1.6. It is important to note that the value of $S_{220}(r)$ determines the averaged orientation of the bananas at radial distance $r$ from the center of the skyrmion as depicted by the right-hand side of Fig. 3b. The radius of a skyrmion $r_s$ can be determined by identifying the location of the first minimum in $S_{220}(r)$, while the depth of this minimum provides information on the twist angle $\theta$ from the center to the border of the skyrmion. In Fig. 3b, we clearly observe that for hexagonally packed skyrmion lattices, the minimum of the orientation correlation function $S_{220}(r)$ shifts to a larger $r$ with decreasing opening angle $\psi$ of the bananas. We plot the radius of the skyrmions $r_s$, as obtained from the minimum of $S_{220}(r)$, as a function of $\psi$ in Fig. 3c, showing clearly that the size of the skyrmions decreases with increasing opening angle $\psi$ as expected. In addition, we represent the particle curvature $R(\psi) = L/\psi$ as a solid line in Fig. 3c. We find that the dependence of the radius of the skyrmion filaments on the opening angle $\psi$ is well-described by the particle curvature $R$, i.e., $r_s(\psi) \sim R(\psi)$.

Furthermore, we can establish a connection between the magnitude of $S_{220}(r)$ and the average angle $\theta$ formed by the skyrmion axis $\hat{\boldsymbol{a}}_\nu$ with the long $\hat{\boldsymbol{u}}$ particle axis of a banana at distance $r$, as indicated on the right-hand $y$-axis of Fig. 3b, by inverting the numerator in Eq. 1. We note that the depths of these minima in $S_{220}(r)$ remain consistent across all investigated opening angles $\psi$ of the banana-shaped particles. The depths of these minima are approximately $S_{220}(r_s) = -0.68$, which corresponds to a tilt angle of 85°, indicating that the formed structures represent half skyrmions. We also observe that the maximum $S_{220}(r) = 1.28$ occurs at $r = 0$, corresponding to an angle of 35° at the center of the skyrmion. This suggests that the banana-shaped particles are already twisted at the center of the skyrmion as can be seen in the snapshots presented in Fig. 1.

## Dynamics of skyrmions

Our simulations at the microscopic particle scale enable us to characterize not only the structure but also the dynamics of the highly structured SL phase (see "methods"). We first examine the translational and orientational motion of the banana-shaped particles. In Fig. 4a, we present the orientational correlation function $C_\theta(t) = \langle \hat{\boldsymbol{u}}(0) \cdot \hat{\boldsymbol{u}}(t) \rangle$ indicating the time scale of the reorientational dynamics exhibited by the banana-like particles. In particular, we find a relaxation time of roughly $\tau_\theta = 10^3 \tau_{\text{MD}}$ with $\tau_{\text{MD}}$ the simulation time unit. Moreover, in Fig. 4b, we present the mean square displacement MSD$(t) = \langle \Delta r^2(t) \rangle$, including its parallel component MSD$_\parallel(t)$ obtained by projecting the displacement vector of each banana onto its initial orientation unit vector $\hat{\boldsymbol{u}}(0)$, as well as its perpendicular MSD$_\perp(t)$ component. At long-time scales, both the MSD and its parallel and perpendicular components increase linearly, indicating diffusive motion characteristics for a fluid-like state. Intriguingly, we observe a crossover between the parallel and perpendicular components of the MSD around time scales close to the characteristic decay time of $C_\theta(t)$. The MSD$_\perp(t)$ shows a smooth transition from short-time to long-time diffusion, while the MSD$_\parallel(t)$ develops an intermediate cage-trapping plateau. Consequently, long-time diffusion in the perpendicular direction surpasses that in the parallel direction. However, at short times $t < \tau_\theta$, the anisotropy in the MSDs demonstrates that the bananas preferentially displace in the direction parallel to their long axis. Considering the

structure of the SL, such observations may indicate a non-trivial structural relaxation of the system. To analyze this, we compute the self-intermediate scattering function $F_s(q,t) = \langle \exp[i\boldsymbol{q} \cdot \Delta \boldsymbol{r}(t)] \rangle$ for varying wave vectors $q\sigma$, as reported in Fig. 4c. Dense systems, such as glasses and supercooled liquids, often exhibit the presence of multiple relaxation times. These relaxation times correspond to distinct processes: a long-time relaxation, which arises from cooperative motion, and a short-time relaxation, which is attributed to the rattling motion of a particle within the transient cage formed by its neighboring particles. Two-step relaxation processes have also been reported in the inter-layer dynamics of dense smectic states[42]. However, we see that our SL is characterized by very fast dynamics with a single structural relaxation decay, typical of dense fluid-like behavior[43]. Finally, we investigate deviations from Gaussian diffusive behavior due to heterogeneous dynamics of "fast" and "slow" particles by measuring the non-Gaussian parameter $\alpha_2(t) = \langle \Delta r^4(t) \rangle / ((1 + 2/3) \langle \Delta r^2(t) \rangle^2) - 1$. Over an extremely broad time scale range, spanning from $10^0 < t/\tau_{MD} < 10^4$, $\alpha_2(t)$ deviates from zero with a peak occurring at approximately $50\tau_{MD}$, see Fig. 4d. This non-Gaussian diffusive behavior arises due to the intricate dynamics exhibited by the individual particles in the highly structured SL. While the overall dynamics resemble that of a simple fluid-like state, the non-Gaussian dynamic heterogeneities highlighted by $\alpha_2(t)$ suggest the existence of particle populations with diverse dynamics closely related to cage rearrangements and cooperative motion in which small clusters of particles move collectively, phenomena commonly observed in various soft-matter systems, including colloidal glasses, crystals, and liquid crystals. Finally, the fluid-like diffusive behavior of the SL is vividly demonstrated in Supplementary Movie 2, showing the rapid diffusion of bananas composing a single skyrmion filament over time.

## Manipulation of skyrmions with electric fields

Skyrmions, as topologically protected and robust field excitations in magnetic systems, have attracted increasing attention for potential spintronic applications[3]. This interest arose from their classification as reconfigurable, ordered matter, offering significant promise for future low-power computer memory that stores information in energetically stable configurations[44]. However, to develop viable technologies, it is necessary to manipulate skyrmions through local heating, external fields, or other techniques[45]. To showcase the robustness of skyrmions formed by banana-like particles in confinement, we perform a series of MD simulations to mimic a prototype system. In this system, an external electric field is employed to either remove or create skyrmions, initially packed within a lattice. Colloidal particles with a dielectric constant mismatch with their surrounding solvent, acquire a dipole moment parallel to the field when subjected to an external electric field. We represent the particles as bananas with a field-induced dipole moment along the long particle axis $\hat{\boldsymbol{u}}$. This allows us to apply an external electric field of strength $E$ to align the particles in the direction parallel to the original skyrmions' axes. By performing cycles alternating between field-on and field-off states, we observe that the system formed of $N = 1000$ particles systematically switches between a uniaxial nematic and a skyrmion phase, as illustrated in Fig. 5 and Supplementary Movie 3.

## Blue phases in bulk

We now explore the possibility for skyrmions to spontaneously emerge in bulk, i.e., in the absence of confining walls, solely due to interparticle interactions. It is well established that in chiral liquid crystals, BPs, a highly dynamic network of chiral skyrmion filaments, may appear in a narrow region between the I and the cholesteric phase N*. To investigate whether or not a BP can also appear near the I to the heliconical $N_{TB}$ phase transition, we consider banana-shaped particles with an opening angle of $\psi = 1.6$, which as explained earlier, yield a direct transition from the I to the $N_{TB}$ phase. However, to

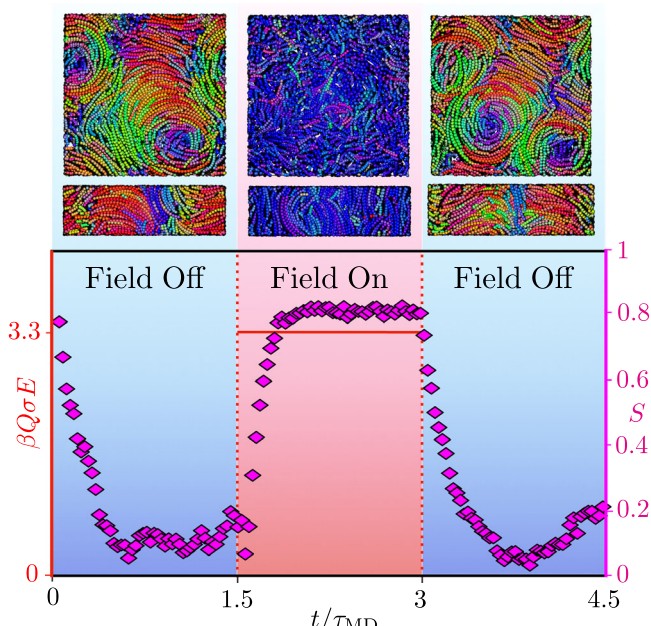

**Fig. 5 | Phase sequence under pulsed electric field.** The red solid line represents the square-wave electric field oscillating with a period of 1.5 $t/\tau_{MD}$ and an amplitude of $\beta Q\sigma E = 3.3$, where $Q$ is the magnitude of the two opposite charges representing the induced dipole moment, and $\sigma$ their separation distance. The nematic order parameter, denoted by $S$ (magenta), closely follows the electric field with a short delay. The banana-like particles, initially in a skyrmion configuration, shift to a nematic configuration with their long particle axes aligned along the electric field. Upon removal of the field, the particles return to their initial skyrmionic state. Source data are provided as a Source Data file.

establish the existence of a BP in this system, large-scale simulations are required, given the long relaxation times and the substantial number of particles involved. For instance, a skyrmion filament with a length comparable to that of a banana comprises ~500 bananas. To reduce the computational costs, we first map out the bulk phase diagram using small and tractable simulations. We conduct simulations on a system comprising $N = 1352$ bananas in the $NPT$ ensemble. We employ a barostat that modifies the system's volume by independently adjusting the box dimensions in the three directions. We plot in Fig. 6 the equation of state, i.e., the pressure $\beta P \sigma^3$ as a function of the packing fraction $\eta$, where $\beta = 1/k_B T$ denotes the inverse temperature and $k_B$ the Boltzmann constant. In addition, we report the corresponding average global values of the nematic order parameter $S$ and the heliconical order parameter $|\mathcal{T}|$. Consistent with the findings in ref. 25, we observe the following phase sequence as the packing fraction $\eta$ increases. At low $\eta$, the system resides in an I phase characterized by low values of both $S$ and $\mathcal{T}$. As $\eta$ increases, the system undergoes a strong first-order phase transition to a chiral $N_{TB}$ phase, as evidenced by discontinuities in $\eta$, $S$, and $|\mathcal{T}|$. At even higher packing fractions, i.e., $\eta > 0.384(1)$, the system exhibits a splay-bend smectic phase $(Sm_{SB})$[25,46].

Our interest lies in the strong first-order I-$N_{TB}$ phase transition with a density gap of $0.282(6) < \eta < 0.324(2)$ as shown in the inset of Fig. 6. We carefully melted a low-density $N_{TB}$ phase with an equilibrium pitch of $\mathcal{P}_{TB} = 34.42\sigma$ at a packing fraction of $\eta = 0.332(9)$, by setting 101 different target pressures variably spaced in the interval $\beta P \sigma^3 \in [0.39, 0.41]$. In addition, we compressed the densest instance of the I phase at a packing fraction of $\eta = 0.282(6)$, to 75 different target pressures in the same pressure range. While no stable skyrmions were observed in the initial configurations, our simulations reveal that the transformation from the I to $N_{TB}$ phase, and vice versa, proceeds via the appearance of intermediate double-twist vortex-like structures.

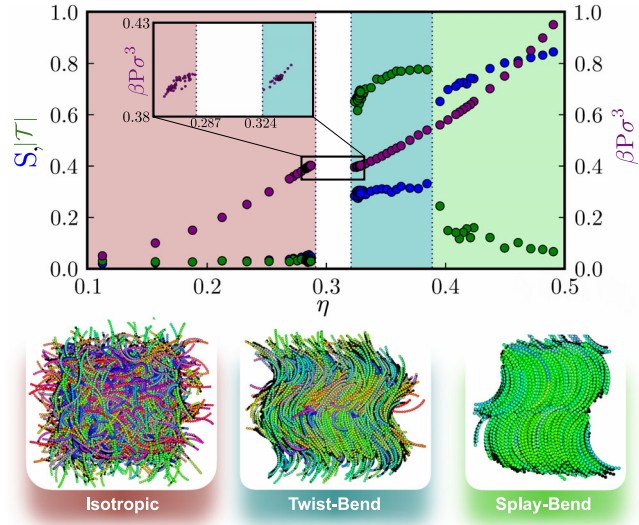

**Fig. 6 | Bulk phase diagram of banana-shaped particles.** The pressure $\beta P \sigma^3$ (purple), nematic order parameter $S$ (blue), and heliconical order parameter $\mathcal{T}$ (green) as a function of the packing fraction $\eta$ of banana-shaped particles with a length-to-diameter ratio $L/\sigma = 16$ and opening angle $\psi = 1.6$ as obtained from MD simulations. The phase diagram displays an isotropic (I) phase at low density, a twist-bend nematic ($N_{TB}$) phase at intermediate density, and a splay-bend smectic ($Sm_{SB}$) phase at high density. Typical configurations are shown in the panels below, where the particles are color-coded according to their orientation. The standard deviation is always at least at the second decimal, except a few measurement of the twist order parameter in the splay-bend phase. The figure with error bars is shown in Supplementary Fig. 4. Source data are provided as a Source Data file.

However, these structures tend to disappear as the system reaches the equilibrium state within $10^5 \tau_{MD}$.

In order to eliminate finite-size effects, which may hinder the formation of skyrmions due to self-interactions through the periodic boundary conditions[47], and encouraged by the appearance of transient skyrmions, we extended our investigation by scaling up the system size to $N = 9000$ banana-shaped particles. We compressed an I configuration to pack fractions beyond the I-$N_{TB}$ phase transition.

We now clearly observe the systematic nucleation of a disordered network of DTCs, consistent with the proposed structure of BPIII. The assembly of DTCs from an I phase occurs within about $10^3 \tau_{MD}$, slightly shorter than the $5 \times 10^3 \tau_{MD}$ to $10^4 \tau_{MD}$ needed to equilibrate a modulated bulk $N_{TB}$ phase with the same number of particles. These time scales can depend on various parameters, such as the initial microscopic configuration and the degree of orientational and positional order in the respective phases. Following the assembly of the skyrmion filaments, the BPIII-like structure remains stable throughout our simulations of $2 \times 10^5 \tau_{MD}$, which is more than two orders of magnitude longer than the measured orientational relaxation time $\tau_\theta$, as shown in Fig. 4. The robustness of this structure over time suggests that the BPIII is thermodynamically stable within an extremely narrow density range. This narrow stability regime is analogous to the way BPs are stable in a small range of temperatures in thermotropic cholesterics.

Our particle-based simulations allow us to characterize the microscopic structure and dynamics of the intricate BPIII phase (see "methods"). To study the emerging patterns and identify the DTCs, we employ the same method described above. Figure 7 offers a visual representation of the phase found in the larger system above the I-$N_{TB}$ transition. By applying our criterion to identify DTCs, we clearly observe the formation of vortex-like structures of flexible, right-handed, and left-handed DTCs. This structure deviates from the typical translational order of the cubic lattices found in BPI and BPII, making it compatible with a BPIII. In contrast to earlier studies[10,37], the frustration

induced by the presence of skyrmions is not resolved by the presence of a network of $\lambda$-lines, but rather by the presence of extended isotropic domains, aligning with previous interpretations of BPIII[11].

In addition, we analyze the structure and dynamics with the methods described above, We observe that the minimum of $S_{220}(r)$ for the BPIII occurs at a slightly lower $r$ compared to the skyrmion lattice (SL) at $\psi = 1.6$, indicating compact and relatively thin skyrmion filaments in BPIII. The depths of these minima are approximately equal to those of the skyrmion lattice, with $S_{220}(r_s) \approx -0.68$, characterizing the filaments as half skyrmions.

Recently, it has been found that in chiral liquid crystals, baby skyrmions form under thin confinement and yield a BPIII in bulk. The skyrmion twist angle and effective diameter differ significantly between the two phases[10]. While confinement results in half-skyrmions, bulk BPIII predominantly features skyrmion filaments with $\sim 35°$ radial twist[10]. In our systems of achiral banana-like particles, we find that the diameter and twist angle of the assembled skyrmions remain practically unchanged under confinement and in bulk. This suggests that the skyrmion size depends similarly on particle curvature in bulk as in confinement. This further emphasizes the relevance of bend and particle curvature in stabilizing skyrmions and constraining their dimensions.

Finally, we measured the intermediate scattering function $F_s(q, t)$, mean square displacement MSD($t$), and the non-Gaussian parameter $\alpha_2(t)$, which are reported in Fig. 4 as a function of time. The results are almost indistinguishable from those for the SL phase reported above, revealing non-Gaussian fluid-like dynamics with a single-step relaxation time.

Our results provide strong evidence for the existence of a novel BPIII composed of bend-stabilized skyrmionic filaments of the nematic orientational field. This phase has not been previously reported in nematic liquid crystals and highlights the importance of our approach to identifying and characterizing DTCs in simulations at the microscopic particle level. The continuous emergence and disappearance of skyrmionic filaments, as well as their fusion and breaking (as shown in Supplementary Movie 4), underscore the complexity and dynamic nature of the skyrmion network in BPIII. The presence of DTCs in a phase introduces a frustration in the continuity of the nematic and polar fields leading to the formation of isotropic domains. Our analysis, in line with the calculations outlined in the Methods section, emphasizes the need of employing multiple approaches to unravel the intriguing behavior that precedes chiral phase transitions. In this context, locally favored DTC configurations introduce frustration into the system, giving rise to the emergence of isotropic domains.

Finally, we note that by systematically exploring the full pressure range of the system, we have found no evidence of stable cubic lattices of DTCs resembling BPI or BPII. This is consistent with the short $N_{TB}$ pitch, which leads to stronger fluctuations that destabilize the crystalline order, as previously argued by Keyes using the Lindemann criterion and the Landau-Peierls argument[48]. Furthermore, our findings are supported by experimental studies by Le et al.[49] and Taushanoff et al.[50], who have also reported the absence of stable crystalline BPs in low-pitch chiral systems. These results highlight the importance of pitch length and its effect on the stability of crystalline BPs in chiral liquid crystals, which has significant implications for the design and engineering of chiral materials for advanced technological applications. However, due to constraints imposed by the accessible length- and time scales of our simulations, as well as our focus on a narrow parameter space (particle length, opening angle, etc.), we cannot definitively exclude the emergence of other types of BPs in systems of banana-like particles.

## Discussion
In conclusion, our observations of 3D BPIII and 2D skyrmion phases are consistent with earlier theoretical predictions[51]. These predictions

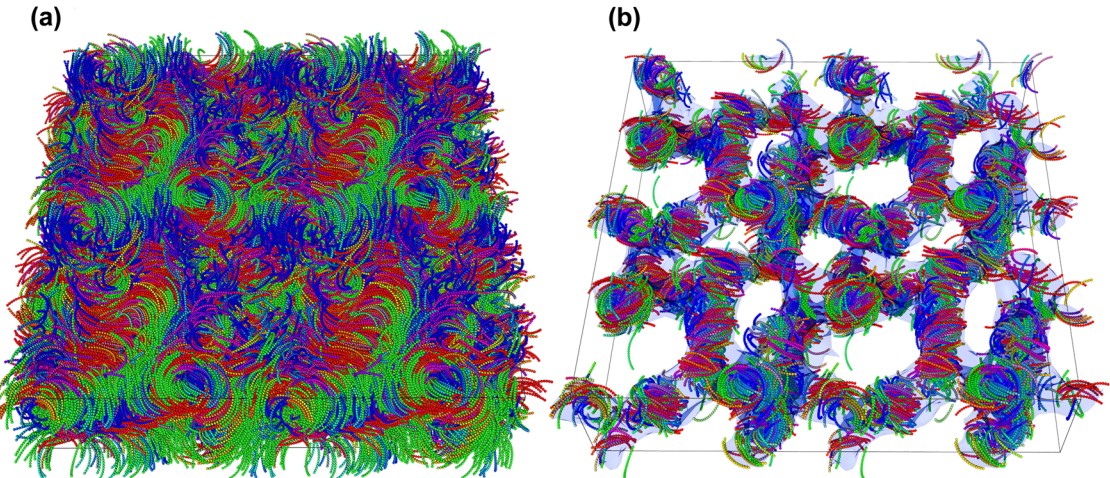

**(a)** **(b)**

**Fig. 7 | Structure of the blue phase III (BPIII) of colloidal banana-shaped particles. a** A typical simulation configuration of BPIII consisting of $N = 9000$ bananas with length-to-diameter ratio $L/\sigma = 16$ and opening angle $\psi = 1.6$ at packing fraction $\eta = 0.33(8)$. **b** To emphasize the characteristic flexible skyrmionic filaments or double-twisted cylinders (DTCs), only bananas with low local nematic order parameter $S < 0.18$ and high heliconal order parameter $|\mathcal{T}| > 0.3$ are shown. The color of the particles is assigned according to their orientation. The isosurface illustrates the enmeshed network of skyrmionic filaments.

showed that the bend-flexoelectric effect, the coupling between polar order and spontaneous bend deformations, could lead to intricate higher-dimensional modulated structures beyond the one-dimensional spatially modulated twist-bend and splay-bend nematic phases. We performed extensive MD simulations on systems of highly curved, achiral hard banana-shaped particles. Our results reveal that the confinement of banana-shaped particles in thin films can lead to the formation of quasi-2D layers containing stable isolated half-skyrmions or high-density hexagonal lattices of half-skyrmions. As the banana-shaped particles themselves are achiral, our simulations reveal a racemic mixture of both right- and left-handed skyrmion filaments. Since the half skyrmion filaments exhibit achirality at the boundary, the resulting arrangement is independent of the handedness of the skyrmion filaments. This differs from recent experimental observations on highly bent molecules (1Cl-N(1,7)-O6) by Kang et al.[52], where a two-dimensional checkerboard arrangement of quarter skyrmions with right-handed and left-handed chiralities was reported. In that case, the chiral arrangement of banana-shaped molecules at the boundary of the skyrmion filaments enforced the observed pattern.

In addition to the established bulk phases of banana-like particles[25], we observe that close to the I-N$_{TB}$ phase boundary, these particles spontaneously self-assemble into a blue phase that structurally resembles a BPIII phase characteristic of cholesteric liquid crystals with a short pitch. This BPIII phase is composed of a highly dynamic network of half-skyrmion filaments embedded in an isotropic background. The dynamics of the BPIII phase are fluid-like but with pronounced non-Gaussian diffusive behavior.

Finally, we have demonstrated that the size of the skyrmions is determined by the curvature of banana-shaped particles. The ability to control the skyrmion size holds significant implications for the production of high-performance displays[53] and the development of electro-optical, optical, and photonic devices[54]. Moreover, we have demonstrated that the skyrmions can be effectively created and annihilated by external electric fields, which not only highlights the robustness of the structures but also underscores their re-configurable and adaptive nature. Our findings highlight that these exotic structures, including skyrmions and the BPIII phase, can be robustly realized in experimental systems composed of achiral bent molecules or colloids, broadening the potential for discovering new phases in diverse materials. We hope that our findings will inspire experimental investigations of bend-stabilized 3D BPIII and 2D skyrmion phases in both thermotropic and lyotropic liquid crystals. Furthermore, it will be

interesting to explore ways to manipulate these skyrmions as well as their mobility for potential applications in future work.

## Methods
### Molecular dynamics simulations
We model each banana-shaped particle as a rigid body consisting of 17 purely repulsive tangent beads of diameter $\sigma$ arranged on an arc segment of $L = 16\sigma$ and radius of curvature $R$, resulting in an opening angle $\psi = L/R$. The individual beads in our system interact with a pseudo hard-sphere potential, which is a cut-and-shifted Mie potential described by

$$u_{\lambda_r,\lambda_a}(r) = \begin{cases} \frac{\lambda_r}{\lambda_r - \lambda_a} \left(\frac{\lambda_r}{\lambda_a}\right)^{\frac{\lambda_a}{\lambda_r - \lambda_a}} \epsilon \left[\left(\frac{\sigma}{r}\right)^{\lambda_r} - \left(\frac{\sigma}{r}\right)^{\lambda_a}\right] + \epsilon & \text{for } r < \sigma \left(\frac{\lambda_r}{\lambda_a}\right)^{\frac{1}{\lambda_r - \lambda_a}} \\ 0 & \text{otherwise ,} \end{cases} \quad (2)$$

where $r$ represents the distance between two beads, and $\epsilon$ and $\sigma$ denote the energy and length scales of the interaction, respectively. By setting $\lambda_r = 50$ and $\lambda_a = 49$ and considering a reduced temperature of $k_B T/\epsilon = 1.5$, we can accurately reproduce the volumetric, structural, and dynamic properties of the discontinuous hard-sphere potential over the entire fluid range[33]. A choice that allows comparison with previous Monte Carlo simulations on hard banana-shaped particles[25].

We investigate the bulk phase behavior of hard banana-shaped particles using Molecular Dynamics (MD) simulations in the *NPT* ensemble, where the number of particles $N$, pressure $P$, and temperature $T$ are fixed. Our simulations are conducted in an orthorhombic simulation box with dimensions $L_x$, $L_y$, and $L_z$, and we apply 3D periodic boundary conditions. The mass $m$ of the beads is taken as the unit of mass, while the units of length, energy, and time are $\sigma$, $\epsilon$, and $\tau_{MD} = \sigma\sqrt{m/\epsilon}$, respectively. We integrate the equations of motion using the velocity Verlet method with a time step of $\delta t = 0.0035\tau_{MD}$. To maintain a temperature of $k_B T/\epsilon = 1.5$, we employ a Langevin thermostat with a damping constant of $0.005\tau_{MD}^{-1}$, and a Berendsen barostat with a coupling constant $200\tau_{MD}$ to keep the average pressure at the desired value $P$. All simulations are carried out using the LAMMPS package[55].

We perform MD simulations on systems of banana-shaped particles with varying sizes, ranging from $N = 1352$ up to $N = 9000$. The systems are initially prepared in either purely isotropic or splay-bend smectic states. To monitor the evolution of the system, we conduct expansion and compression simulations and track various

observables, including packing fraction, temperature, various order parameters, and convergence to the target pressure.

We find that the equilibration time of the system depends on its size. For larger systems, achieving equilibration may require up to $10^8$ MD steps.

To investigate the behavior of banana-shaped particles under thin confinement, we perform MD simulations in the presence of two planar structureless walls, whose surface normals are aligned to the $z$-axis. The walls are modeled by truncated and shifted harmonic potentials applied to the $z$-coordinates of individual beads. The potentials are defined as

$$u_{w,1}(z) = \begin{cases} \epsilon_w (z - z_1)^2 & \text{for } z < z_1 \\ 0 & \text{otherwise} \end{cases} \qquad (3)$$

and

$$u_{w,2}(z) = \begin{cases} \epsilon_w (z - z_2)^2 & \text{for } z > z_2 \\ 0 & \text{otherwise ,} \end{cases} \qquad (4)$$

where $\epsilon_w = 0.1 k_B T / \sigma^2$ is the strength of the wall-particle interaction, $z_1$ and $z_2 > z_1$ denote the location of the "lower" and "upper" wall, respectively. The initial configurations consist of a low-density monolayer of $N = 4500$ particles in an isotropic state placed between the two confining walls separated by a distance $H^{st} \equiv z_2 - z_1$. The systems are then slowly laterally compressed at pressure $\beta P_{xx}\sigma^3 = \beta P_{yy}\sigma^3 = 0.05$ until the desired effective packing fraction is reached. The effective packing fraction is approximated as $\eta = N v_p / V^{eff}$, where $v_p = \pi \sigma^2 / 4 ((2/3)\sigma + L)$ is the approximate volume of a banana-shaped particle, and $V^{eff} = L_x L_y H$ is the effective volume in which the particles are confined. Because of the softness of the confining walls, the system reaches an equilibrium effective confinement length at long times, $H > H^{st}$. This length is determined by the points $z_l < z_1$ and $z_t > z_2$ at which the one-dimensional density profiles of individual particle beads $\rho(z)$ fall to zero. After achieving the desired packing fraction of $\eta \approx 0.33$, which corresponds to the BPIII configuration in bulk, we further equilibrate the system in the canonical ensemble with fixed $N$, $V$, and $T$ for up to $10^8$ MD time steps.

## Order parameters

The nematic director field of the $N_{TB}$ phase is parametrized by the integral curve $\hat{n} = (\sin \gamma \cos hz, \sin \gamma \sin hz, \cos \gamma)$, with $h = 2\pi / \mathcal{P}_{TB}$ the wavenumber of the twist-bend modulation[25,51]. We further assume[25] that the polarization vector field is perpendicular to $\hat{n}$ in the direction of the higher bend $\hat{p} = \hat{n} \times (\nabla \times \hat{n}) / ||\hat{n} \times (\nabla \times \hat{n})|| = (-\sin \gamma \cos hz, \sin \gamma \sin hz, 0)$, to form a local, right-handed ortho-normal frame at every point we also define $\hat{m} = \hat{n} \times \hat{p} = (-\sin \gamma \sin hz, -\sin \gamma \cos hz, \cos \gamma)$.

Xu and Chen have demonstrated that the Landau-de Gennes free energy of a nematic phase comprising of particles with $C_{2V}$ symmetry, such as bananas, can be expanded up to second order in terms of two symmetric traceless order parameters[56]: the well-known nematic order parameter $\mathbf{Q} = \langle \frac{3}{2}\hat{n} \otimes \hat{n} - \frac{1}{2}\mathbb{I} \rangle$, and $\mathbf{M}_2 = \langle \hat{n} \otimes \hat{m} + \hat{m} \otimes \hat{n} \rangle$, where $\langle ... \rangle = \frac{1}{\mathcal{P}_{TB}} \int_0^{\mathcal{P}_{TB}} ... dz$ denotes the integral average over $z$ over one full period. By substituting the expressions for $\hat{n}$ and $\hat{m}$ (without the need of diagonalizing the tensors given the choice of reference frame), we obtain $\mathbf{M}_2 = (\sin 2\gamma) \, \text{Diag}(1, -1/2, -1/2)$. We thus find that for a cholesteric phase with $\gamma = 0$, the heliconical order parameter $\mathcal{T}$ is zero. On the other hand, the eigenvalues are maximal at $\gamma = \pm \pi/4$, with the heliconical order parameter represented by the largest eigenvalue in absolute value, $\mathcal{T} = \pm 1$. Remarkably, we note that the sign gives the handedness of the phase. Substituting $\gamma = \pm \pi/4$ back into $\mathbf{Q} = \text{Diag}(\frac{1}{4}(1 + 3\cos 2\theta), \frac{1}{4}(-2 + 3\sin^2\theta), \frac{1}{4}(-2 + 3\sin^2\theta))$, we find that its highest eigenvalue, regardless of the sign of $\gamma$, is $S = 1/4$. These values are in full agreement with the equation of state presented in Fig. 6. The slight discrepancy with a value of $|\mathcal{T}| = 1$ can likely be attributed to a slightly lower conical angle or a mild deviation from a pure $N_{TB}$ phase. It is worth comparing these findings with recent work by Rossetto and Selinger, which reveals that the saddle-splay mode of the Frank-Oseen free energy for modulated phases may be accompanied by the emergence of an octupolar order parameter[57]. This octupolar order parameter can be decomposed as the sum of $\mathbf{M}_2$ and the commonly used biaxial order parameter $\mathbf{M}_1 = \langle \hat{p} \otimes \hat{p} - \hat{m} \otimes \hat{m} \rangle$.

## Vortex detection

Previous work have employed Landau-de Gennes theories to study the stability and structure of BPs. To visualize the intricate details of the skyrmion networks, nematic order parameter thresholds were defined based on the varying local orientation from the center to the periphery of the skyrmions. Henrich et al.[37] employed order parameter iso-surfaces at a value $S = 0.12$ to visualize the skyrmions, whereas Pišljar et al.[10] used a value $S = 0.14$. However, in particle-based simulations, an extension of the above-mentioned methodology is required to detect and extract the skyrmions from the full configurations. To achieve this, we compute the local tensor order parameters $\mathbf{Q}$ and $\mathbf{M}_2$ for each banana. This calculation is performed over all neighbors within a radial distance $r_S$ and $r_{\mathcal{T}}$ centered around the center-of-mass of a banana, for $\mathbf{Q}$ and $\mathbf{M}_2$, respectively. Subsequently, we calculate the eigenvalues $S$ and $\mathcal{T}$ of the computed $\mathbf{Q}$ and $\mathbf{M}_2$ tensors. To visualize the skyrmions, we apply threshold values for $S$ and $\mathcal{T}$. By setting a threshold of $S < 0.18$, we can successfully identify all skyrmions, along with some individual bananas present in the box. To further refine the identification of skyrmions and remove the individual bananas, we impose the condition $|\mathcal{T}| > 0.3$. Once the skyrmions are identified, we compute the center-of-mass and axis of the skyrmion filament. The axis is determined by the eigenvector corresponding to the highest eigenvalue (in absolute value) $\mathcal{T}$ (see Supplementary Fig. 2). In addition, we quantify the handedness of the skyrmions by considering the sign of $\mathcal{T}$ as shown in Supplementary Fig. 1.

## Orientation correlation functions

To determine the radius and the order of the skyrmions we rely on orientational correlation functions. Supplementary Fig. 3 shows that the $N_{TB}$ phase exhibits both odd and even banana-banana orientational correlations[41], signaling the chiral symmetry breaking, whereas only even banana-banana orientational correlations emerge in skyrmion phases, emphasizing its racemic nature. To infer the skyrmion radius, we introduce the banana-skyrmion orientational correlation function in Eq. (1). To characterize the spatial arrangement of bananas in a skyrmion filament, we employ a modified correlation function that takes into account the center of mass and orientation of the skyrmion filament as described above. This approach differs from the standard pair correlation functions, as it involves summing over all pairs of banana-skyrmion combinations within a given configuration. The distance between a banana and a skyrmion is measured as the orthogonal distance from the banana to the skyrmion axis. To ensure that the correlation function captures purely orientational information and avoids any artificial density modulation caused by particle orientations in cylindrical slabs around the skyrmion axis, we normalize it with $S_{000}$. This normalization factor essentially represents the banana-skyrmion radial distribution function, enabling us to focus exclusively on the orientational aspects of the skyrmion.

## Dynamics

To investigate the translational dynamics of the banana-like particles in the BPIII phase, we calculated the mean square displacement, which reads

$$\langle \Delta r^2(t) \rangle = \frac{1}{N} \left\langle \sum_{i=1}^{N} \left[ \mathbf{r}_i(t) - \mathbf{r}_i(0) \right]^2 \right\rangle, \qquad (5)$$

where $N$ is the number of particles, $\boldsymbol{r}_i(t)$ the position of the center-of-mass of banana $i$ at time $t$ and $\langle \cdots \rangle$ denotes an ensemble average. The anisotropy in the translational motion of the particles is characterized by computing the components parallel and perpendicular to the initial orientation of the individual particles. In particular, the banana displacement vector is projected onto its initial orientation unit vector $\hat{\boldsymbol{u}}(0)$[58].

Dynamic heterogeneities are assessed by using the non-Gaussian parameter defined as

$$\alpha_2(t) = \frac{\langle \Delta r^4(t) \rangle}{(1 + 2/d)\langle \Delta r^2(t) \rangle^2} - 1, \tag{6}$$

where $d = 3$ is the dimensionality of the system. To provide a quantitative measure of the time associated with the structural relaxation of the system and to quantify the decay of their density fluctuations, we calculate the self-intermediate scattering function, which reads

$$F_s(q,t) = \frac{1}{N}\left\langle \sum_{i=1}^{N} \exp\left(i\boldsymbol{q} \cdot \left[\boldsymbol{r}_i(t) - \boldsymbol{r}_i(0)\right]\right) \right\rangle, \tag{7}$$

with $\boldsymbol{q}$ the wave vector.

Finally, to quantify the rotational dynamics, we compute the so-called orientational correlation function

$$C_\theta(t) = \frac{\langle P_1\left[\hat{\boldsymbol{u}}(t) \cdot \hat{\boldsymbol{u}}(0)\right] \rangle}{\langle u^2 \rangle}, \tag{8}$$

where $P_1$ is the first-order Legendre polynomial. From the full time correlation function, an orientational relaxation time can be estimated as $\tau_\theta = \int_0^\infty C_\theta(t)dt$. Here, we approximate it as the time $t = \tau_\theta$ at which $C_\theta(t) = (1/e)$.

## Data availability
Source data files are available with the paper. Source data are provided with this paper.

## Code availability
Simulations were performed with an open-source package as referenced in the manuscript. Analysis codes can be made available from the authors upon request.

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

## Acknowledgements

R.S., G.C.-V., and M.D. thank Susana Marin-Aguilar, Fabrizio Camerin, Bela Mulder, and René van Roij for useful discussions that helped shape this work. R.S. acknowledges financial support from the Netherlands Organization for Scientific Research (NWO) (Grant No. OCENW.K-LEIN.423) and G.C.-V. for the NWO ENW PPS Fund 2018-Technology Area Soft Advanced Materials (Grant No. ENPPS.TA.018.002). This work was also supported by the European Research Council (Grant No. ERC-2019-ADV-H2020 884902).

## Author contributions

These authors contributed equally: R.S., G.C.-V., and M.D. conceived and designed the project. R.S. and G.C.-V. performed the numerical simulations and analysis. R.S. and G.C.-V. wrote the original draft of the paper. All authors participated in the discussions, and reviewed and edited the manuscript.

## Competing interests

The authors declare no competing interests.
