## [Peer Review File · Nature Communications]

REVIEWER COMMENTS

Reviewer #1 (Remarks to the Author):

This work summarizes the computational finding of skyrmions and blue phases in systems of banana-shaped particles with hard-like type of interactions. The novelty of these findings lies on the fact that chiral phases formed by skyrmions are stable in systems of achiral particles. The authors study the phase behaviour of these particles under confinement, for which they find (lattices of) skyrmions, and in the absence of confinement, a potential BPIII. Finally, they also show how the size of the skyrmions can be tuned by the opening angle of the particles.

This is a very nicely written and well-structured letter which represents an advance in the liquid crystal fields of banana-shaped building blocks. I believe this work will inspire new experimental and theoretical work on achiral molecules or colloids. In summary, the novelty and quality of this work are such that I would recommend its publication in Nature Communications. Nonetheless, there are few points I would like the authors to address before its final publication:

1. First of all, the authors should clarify in the introduction that in ref. 10, the microscopic structure of the BPIII, is in fact, discussed in detail, and also as a function of confinement. The sentence “BPIII is mysterious in terms of its microscopic structure” might lead to some confusion in this regard. While it is indeed nice to have further work confirming its microscopic details, it is not that it has not been unveiled at this point. In my opinion, the great novelty of the current work is not the unveiling of the structure of the BPIII (as it was already discussed in ref 10), but the fact that this is found in a system of achiral particles (and not a system with high chirality, as is ref 10).

2. My second point is about the bend-flexoelectric effect. If I understand correctly, flexoelectricity is the coupling between strain gradients and electric polarization of the molecules. In systems of molecular banana-shaped particles, the bent shape of the molecules gives them, indeed, a net dipolar moment perpendicular to the long axis of the

molecules, which in turn gives the phase macroscopic polarization (even in the absence of an electric field). On the other hand, the system discussed here, is formed by a collection of hard particles, where an electric polarization can only be induced in the presence of an electric field (and this not really polar, as it is parallel to the particles' long axis). As such, I think the authors should clarify the use of 'bend-flexoelectricity' in the text, as there is no polarization of the phase here; is just the 'polar' axis of the particles presents some order. Perhaps 'steric bend-flexoelectricity' or 'bend-flexostericity' could be a better term? I guess the key idea here is to not misguide the reader, and clarify that there is no electric polarization but steric polarization.

3. In page 5. "We plot the radius of the skyrmions ... showing clearly that the size of the skyrmions decreases with increasing particle curvature as expected". If I understand correctly, the length of the particles is kept constant in these simulations, which means that for increasing opening angle, the curvature of the particle decreases ($\theta = 1/R$). So is it possible that the correct version is that the size of the skyrmions decreases with increasing opening angle or decreasing radius of curvature?

4. Could you elaborate a bit on the conclusion that $S_{220}(r_s) = -0.68$ corresponds to a half skyrmion? Also, would be any way the particles in the right-hand axis of Fig. 3b could be colored the same way they are in the skyrmion inside the plot?

5. The authors mention using 'extensive' particle-based simulations, and discuss them very nicely in page 8-10. It would be helpful to compare the time scales needed to observe the BP_{III}, compared to the timescale needed to observe the N_{SB}. Especially in the paragraph where the entire duration of the simulations $2 \times 10^5 \mu\text{s}$ is mentioned.

6. In the same way that the opening angle is explored in the confined experiments, how do you expect the role of opening angle will be the same in the bulk experiments? While I understand the computational time to study different shapes might be too long, could the authors comment on their expectations?

7. Finally, regarding the end of the final paragraph of the letter (before the methods). "We anticipate that the BP_{III}, as explored in this study, can also be observed in thermotropic

liquid crystals of bent-core mesogens.” Again, in ref.10, a BPIII has been found in a system of thermotropic LC (mixtures of bent-core and chiral molecules). Which brings me back to point 1, I think this is very inspiring work, not for necessarily unveiling the structure of the BPIII, but to show that these exotic structures can be observed in system of achiral molecules/colloids.

Minor details:

1. I might have missed it, but could it be possible to add a color legend for the orientation field of the particles in 3D (at least in Figure 1)? It is obvious for the images that purple is perpendicular to the x-y plane and green parallel, but it would be even better if the legend could be found somewhere.

2. In page 3. ‘As will be discussed next, such phase is reminiscent of... cholesteric liquid crystals’. Here it misses a reference, probably 10 or 17?. Same after the first sentence of ‘manipulation of Skyrmions with Electric Fields” regarding the spintronic applications.

3. Perhaps a clarification that the manipulation of skyrmions with electric field is in the system with confinement. e.g. ‘To showcase the robustness of skyrmions formed by banana-shaped particles in confinement,..”

Reviewer #3 (Remarks to the Author):

The authors describe a numerical study of the different mesophases formed by achiral bent-core particles. The results demonstrate that hard-core banana shaped particles spontaneously form chiral structures, such as skyrmions or double twist cylinders. Using molecular dynamic simulations, the phase behaviour of these particles is studied in a tight confinement. Fixing the particle size, opening angle (curvature) and a packing fraction, a phase sequence of Splay-Bend (SB), Isolated Skyrmion (IS), Skyrmion lattice (IL) and Blue

Phase III (BPIII), is observed in the simulations when the gap-size between the confining wall is increased. It was shown that the skyrmions can be manipulated by an electric field and in the SL phase, the size of the skyrmions was studied by calculating orientation correlation function. It was shown that the size decreases with increasing opening angle ψ of the banana shaped particle. Finally, the possibility to observe BPIII in bulk samples is studied. To do this, the authors target the Isotropic - Twist Bend nematic transition point in the phase diagram, which is known from previous work (ref 26), and reproduced in this article using MD simulations (Fig. 6). At small system size (~ 1300 banana "molecules") a transient double-twist is observed. Increasing the system size to ~ 9000 particles, a disordered network of double twist cylinders is observed to be stable within the simulation time. This suggests that BPIII might have a narrow thermodynamic stability window in the phase diagram, however a possibility of metastability was not ruled out.

The article is well written and clearly presented. I am not an expert in skyrmions, banana shaped particles, nor methods presented in this article, but I do work periodically with liquid crystals and simulations. I found the results clear and the fact very interesting, that these structures can arise due to spontaneous chiral symmetry breaking in the system. Generally, the results are convincing and clearly presented. I find all this very much inline with publication in Nature Communication.

As mentioned above, I am not familiar enough with the details of the field to really judge the general, (or experimental) relevance of the observed results. I do have some worries concerning, how surprising nor how experimentally relevant these results are. I think clarifying this further, if possible, would render the results more accessible to a rather general reader, like myself.

Further questions/comments

1) Could the electric field used to make skyrmions mobile? I think this would certainly increase the impact of the article.

2) I do not quite understand the isolated skyrmion phase. Is it really just one, or would this depend on the size of the system? In which case it might be some kind of isotropic skyrmion state?

3) I am bit confused with the achirality of the observed states. I guess one could have either, a collapse of chirality such that all the twists within the system have the same handedness, while on averaging over all the samples, a racemic results is recovered. Another, way, which I think is the case here, is to have a racemic state where half the twists are right and other half are left handed. However, looking at the figure S1 in the supplement, shows non-racemic handedness of the skyrmion filaments.

Let's twist again: Achiral Hard Bananas Assemble Double-Twist Skyrmions and Blue Phases

by Rodolfo Subert, Gerardo Campos-Villalobos and Marjolein Dijkstra

Dear Reviewer #1 and Reviewer #3,

Thank you very much for your reviewing effort and the comments on our manuscript "Let's twist again: Achiral Hard Bananas Assemble Double-Twist Skyrmions and Blue Phases" NCOMMS-24-06118.

We appreciated the careful and in depth reading, your valuable comments, and the excellent suggestions, which have greatly contributed to the improvement of our work. Below, we address the concerns raised point by point.

Yours Sincerely,

Rodolfo Subert, Gerardo Campos-Villalobos and Marjolein Dijkstra

• Reviewer 1

This work summarizes the computational finding of skyrmions and blue phases in systems of banana-shaped particles with hard-like type of interactions. The novelty of these findings lies on the fact that chiral phases formed by skyrmions are stable in systems of achiral particles. The authors study the phase behaviour of these particles under confinement, for which they find (lattices of) skyrmions, and in the absence of confinement, a potential BPIII. Finally, they also show how the size of the skyrmions can be tuned by the opening angle of the particles.

This is a very nicely written and well-structured letter which represents an advance in the liquid crystal fields of banana-shaped building blocks. I believe this work will inspire new experimental and theoretical work on achiral molecules or colloids. In summary, the novelty and quality of this work are such that I would recommend its publication in Nature Communications.

We thank the reviewer for their positive comments and for recommending our work for publication in Nature Communications.

Nonetheless, there are few points I would like the authors to address before its final publication:

- 1. First of all, the authors should clarify in the introduction that in ref. 10, the microscopic structure of the BPIII, is in fact, discussed in detail, and also as a function of confinement. The sentence "BPIII is mysterious in terms of its microscopic structure" might lead to some confusion in this regard. While it is indeed nice to have further work confirming its microscopic details, it is not that it has not been unveiled at this point. In my opinion, the great novelty of the current work is not the unveiling of the structure of the BPIII (as it was already discussed in ref 10), but the fact that this is found in a system of achiral particles (and not a system with high chirality, as is ref 10).**

We fully agree with the reviewer that the primary novelty of our work lies in the self-assembly of skyrmions in a system of achiral particles, rather than in the characterisation of the structure of the bulk BPIII. Our intention with this sentence was to highlight the challenges in unraveling the structure of such a phase due to its non-periodic nature, which contrasts sharply with its crystalline counterparts (BPI and BPII). To avoid any potential misinterpretation of the novelty of our work, we have modified this sentence.

We reformulated the paragraph as follows:

BPI and BPII consist of rigid DTCs arranged in body-centered cubic and simple cubic lattices, respectively, intertwined with a lattice of topological defect lines. The structure of the third blue phase, BPIII, posed more difficulties and was only recently unveiled [7], establishing it as a liquid tangle of DTCs or skyrmion filaments. This finding explains its hazy appearance, which has led to its alternative name "blue fog". The third blue phase BPIII, also called the "blue fog" because of its hazy appearance, has posed enduring mysteries regarding its microscopic structure, dynamics, and stability. Recent experimental findings reinforce the notion that BPIII corresponds to a liquid tangle of DTCs or skyrmion filaments[2]. Direct observation of DTCs within bulk BPIII is challenging due to the small size of the cylinders and the strong thermal fluctuations in this phase.

2. My second point is about the bend-flexoelectric effect. If I understand correctly, flexoelectricity is the coupling between strain gradients and electric polarization of the molecules. In systems of molecular banana-shaped particles, the bent shape of the molecules gives them, indeed, a net dipolar moment perpendicular to the long axis of the molecules, which in turn gives the phase macroscopic polarization (even in the absence of an electric field). On the other hand, the system discussed here, is formed by a collection of hard particles, where an electric polarization can only be induced in the presence of an electric field (and this not really polar, as it is parallel to the particles' long axis). As such, I think the authors should clarify the use of 'bend-flexoelectricity' in the text, as there is no polarization of the phase here; is just the 'polar' axis of the particles presents some order. Perhaps 'steric bend-flexoelectricity' or 'bend-flexostericity' could be a better term? I guess the key idea here is to not misguide the reader, and clarify that there is no electric polarization but steric polarization.

We acknowledge the reviewer's concern regarding the term *flexoelectric*, given that our model particles lack electrostatic character compared to banana-shaped molecules. However, this term aligns with recent literature on modulated nematic liquid crystals [1, 3–6], where it refers to the general coupling phenomenon where bend or splay induces net polarization. While terms like "steric bend-flexoelectricity" or "bend-flexostericity" may be more precise, they have not been previously introduced. Therefore, we prefer to maintain our original choice but now clarify more precisely its convention in defining the "bend-flexoelectric effect" to prevent misinterpretations.

The emergence of spatial modulations in the nematic director field of the N_{TB} phase is attributed to the bend-flexoelectric effect, which arises from the coupling between the polar order of the particles and the spontaneous bend deformations arising from the curvature of the bananas in which spontaneous bend deformations induce a net polarization. The polar order may be electrostatic, magnetic, or steric ordering of molecular or particle shape [3]. To resolve the problem that pure bend deformations cannot uniformly fill three-dimensional space...

3. In page 5. "We plot the radius of the skyrmions ... showing clearly that the size of the skyrmions decreases with increasing particle curvature as expected". If I understand correctly, the length of the particles is kept constant in these simulations, which means that for increasing opening angle, the curvature of the particle decreases ($\theta = 1/R$). So is it possible that the correct version is that the size of the skyrmions decreases with increasing opening angle or decreasing radius of curvature?

We thank the reviewer for this important observation, which has enabled us to rectify this oversight. Indeed the skyrmion radius does not decrease but increases with increasing particle curvature, and conversely decreases with increasing opening angle.

The sentence has been corrected:

We plot the radius of the skyrmions r_s , as obtained from the minimum of $S_{220}(r)$, as a function of ψ in Fig. 3(c), showing clearly that the size of the skyrmions decreases with increasing particle curvature with increasing opening angle ψ as expected. In addition, we represent the particle curvature $R(\psi) = L/\psi$ as a solid line in Fig. 3(c). We find that the dependence of the radius of the skyrmion filaments on the opening angle ψ is well-described by the particle curvature R , i.e. $r_s(\psi) \sim R(\psi)$.

4. Could you elaborate a bit on the conclusion that $S_{220}(r_s) = -0.68$ corresponds to a half skyrmion? Also, would be any way the particles in the right-hand axis of Fig. 3b could be colored the same way they are in the skyrmion inside the plot?

We appreciate the reviewer's suggestions. We agree with the referee that it would be beneficial to clarify how the tilt angle of the long axis of the particles relative to the skyrmion central axis is related to the value of Eq. 1. Additionally, to facilitate a more direct understanding, we also concur that colouring the particles in Fig. 3b based on their orientation in space (and thus matching the colors assumed in the skyrmions) would be helpful.

We have coloured the particles accordingly. Additionally, the pixel luminescence parameter Γ has been adjusted to ensure matching colouring in all skyrmions visualizations. The caption has been revised:

The particles on the right-hand axis of the plot are coloured according to a rotation around the \hat{x} -axis. The top particle is parallel to \hat{z} , the bottom one is aligned parallel to \hat{y} .

The paragraph on $S_{220}(r_s)$ has been reformulated:

Furthermore, we can establish a connection between the magnitude of $S_{220}(r)$ and the average angle θ formed by the skyrmion axis \hat{a}_s , with the long \hat{u} particle axis of a banana at distance r , as indicated on the right-hand y-axis of Fig.3(b), by inverting the numerator in Eq. 1. ~~we~~ We note that the depths of these minima in $S_{220}(r)$ remain consistent across all investigated opening angles ψ of the banana-shaped particles. The depths of these minima are approximately $S_{220}(r_s) = -0.68$, which corresponds to a tilt angle of 85° , indicating that the formed structures represent half skyrmions.

5. **The authors mention using ‘extensive’ particle-based simulations, and discuss them very nicely in page 8-10. It would be helpful to compare the time scales needed to observe the BPIII, compared to the timescale needed to observe the N_{SB} . Especially in the paragraph where the entire duration of the simulations $2 \times 10^5 \tau$ is mentioned.**

We thank the reviewer for raising this important point. The observation of the N_{SB} phase occurs only under very thin confinement, where the bananas must align with the confining walls, thereby hindering the nucleation of both the N_{TB} or the SL phase. The N_{SB} nucleation time is thus unrelated to that of the bulk BPIII. Nevertheless, we agree on the relevance of providing at least a rough comparison on time scales needed to observe the assembly of the bulk BPIII and the N_{TB} phase. Such a comparison offers insights into the length of the simulations, which is crucial for reproducibility purposes. However, we also emphasize that the nucleation speed of each phase may depend on various parameters. These include the initial and final thermodynamic macroscopic constraints, the degree of orientational and positional order in the involved phases, initial microscopic configuration, and the shape and size of the box, which, if not commensurate with the N_{TB} pitch, can slow down nucleation.

We have revised the sentence to include a comparison on the characteristic time scales needed to observe the assembly of the bulk BPIII and N_{TB} phase:

We now clearly observe the systematic nucleation of a disordered network of DTCs, consistent with the proposed structure of BPIII. The assembly of DTCs from an I phase occurs within about $10^3 \tau_{MD}$, slightly shorter than the $5 \times 10^3 \tau_{MD}$ to $10^4 \tau_{MD}$ needed to equilibrate a modulated bulk N_{TB} phase with the same number of particles. These time scales can depend on various parameters, such as the initial microscopic configuration and the degree of orientational and positional order in the respective phases. Following the assembly of the skyrmion filaments, the BPIII-like ~~This~~ structure remains stable throughout our simulations of $2 \times 10^5 \tau_{MD}$, which is more than two orders of magnitude longer than the measured orientational relaxation time τ_θ , as shown in Fig. 4. The robustness of this structure over time suggests that the BPIII is thermodynamically stable within an extremely narrow density range. This narrow stability regime is analogous to the way BPs are stable in a small range of temperatures in thermotropic cholesterics.

6. **In the same way that the opening angle is explored in the confined experiments, how do you expect the role of opening angle will be the same in the bulk experiments? While I understand the computational time to study different shapes might be too long, could the authors comment on their expectations?**

We thank the referee for the relevant comment. Our observations of banana-shaped particles in bulk for a single opening angle show that the diameter and the twist angle of skyrmions in the BPIII and SL phase are very similar, unlike in experiments with chiral particles. This difference further emphasizes the relevance of bend and particle curvature in stabilizing the skyrmions and constraining their dimensions.

We expanded the paragraph as follows:

Recently, it has been found that in chiral liquid crystals, baby skyrmions form under thin confinement and yield a BPIII in bulk. The skyrmion twist angle and effective diameter differ significantly between the two phases [9]. While confinement results in half-skyrmions, bulk BPIII predominantly features skyrmion filaments with approximately 35° radial twist [9]. In our systems of achiral banana-like particles, we find that the diameter and twist angle of the assembled skyrmions remain practically unchanged under confinement and in bulk. ~~This suggests that the skyrmion size depends similarly on particle curvature in bulk as in confinement. This further emphasizes the relevance of bend and particle curvature in stabilizing skyrmions and constraining their dimensions.~~

7. **Finally, regarding the end of the final paragraph of the letter (before the methods). “We anticipate that the BPIII, as explored in this study, can also be observed in thermotropic liquid crystals of bent-core mesogens.” Again, in ref.10, a BPIII has been found in a system of thermotropic LC (mixtures of bent-core and chiral molecules). Which brings me back to point 1, I think this is very inspiring work, not for necessarily unveiling the structure of the BPIII, but to show that these exotic structures can be observed in system of achiral molecules/colloids.**

We agree with the reviewer that the key message of our work is that these exotic structures can be observed in system of achiral molecules or colloids. We have modified this sentence according to the reviewer’s suggestion.

Our findings highlight that these exotic structures, including skyrmions and the BPIII phase, can be robustly realized in experimental systems composed of achiral bent molecules or colloids, broadening the potential for discovering new phases in diverse materials. ~~We anticipate that the BPIII phase, as explored in this study, can also be observed in thermotropic liquid crystals of bent-core mesogens, expanding the scope beyond lyotropic liquid crystals examined here.~~ We hope that our findings will inspire experimental investigations of bend-stabilized 3D BPIII and 2D skyrmion phases in both thermotropic and lyotropic liquid crystals.

Minor details:

1. **I might have missed it, but could it be possible to add a color legend for the orientation field of the particles in 3D (at least in Figure 1)? It is obvious for the images that purple is perpendicular to the x-y plane and green parallel, but it would be even better if the legend could be found somewhere.**

We agree with the reviewer that adding a color legend would enhance the clarity of the plots.

We have added a color legend in Figure 1, clarifying the color coding of the particles. The particles are colored according to the orientation of their long \hat{u} particle axis using the RGB color coding displayed in Fig. 1b. The caption of Figure 1 has been modified to include the legend description.

(b) Color legend for banana-shaped particles. The particles in the displayed configurations are colored according to the orientation of their long \hat{u} particle axis using the RGB color coding as shown in the legend.

2. **In page 3. ‘As will be discussed next, such phase is reminiscent of... cholesteric liquid crystals’. Here it misses a reference, probably 10 or 17?. Same after the first sentence of ‘manipulation of Skyrmions with Electric Fields’ regarding the spintronic applications.**

We have added the suggested references and reference 4 for the spintronic applications.

3. **Perhaps a clarification that the manipulation of skyrmions with electric field is in the system with confinement. e.g. ‘To showcase the robustness of skyrmions formed by banana-shaped particles in confinement,..’**

We agree with the reviewer that it is important to emphasize that the manipulation of skyrmions is investigated in a system under confinement.

To showcase the robustness of skyrmions formed by banana-like particles in confinement, we perform a series of MD simulations to mimic a prototype system. In this system, an external electric field is employed to either remove or create skyrmions, initially packed within a lattice.

• Reviewer 3

The authors describe a numerical study of the different mesophases formed by achiral bent-core particles. The results demonstrate that hard-core banana shaped particles spontaneously form chiral structures, such as skyrmions or double twist cylinders. Using molecular dynamic simulations, the phase behaviour of these particles is studied in a tight confinement. Fixing the particle size, opening angle (curvature) and a packing fraction, a phase sequence of Splay-Bend (SB), Isolated Skyrmion (IS), Skyrmion lattice (IL) and Blue Phase III (BPIII), is observed in the simulations when the gap-size between the confining wall is increased. It was shown that the skyrmions can be manipulated by an electric field and in the SL phase, the size of the skyrmions was studied by calculating orientation correlation function. It was shown that the size decreases with increasing opening angle ψ of the banana shaped particle. Finally, the possibility to observe BPIII in bulk samples is studied. To do this, the authors target the Isotropic - Twist Bend nematic transition point in the phase diagram, which is known from previous work (Ref.[1]), and reproduced in this article using MD simulations (Fig. 6). At small system size (1300 banana "molecules") a transient double-twist is observed. Increasing the system size to 9000 particles, a disordered network of double twist cylinders is observed to be stable within the simulation time. This suggests that BPIII might have a narrow thermodynamic stability window in the phase diagram, however a possibility of metastability was not ruled out.

The article is well written and clearly presented. I am not an expert in skyrmions, banana shaped particles, nor methods presented in this article, but I do work periodically with liquid crystals and simulations. I found the results clear and the fact very interesting, that these structures can arise due to spontaneous chiral symmetry breaking in the system. Generally, the results are convincing and clearly presented. I find all this very much inline with publication in Nature Communication.

As mentioned above, I am not familiar enough with the details of the field to really judge the general, (or experimental) relevance of the observed results. I do have some worries concerning, how surprising nor how experimentally relevant these results are. I think clarifying this further, if possible, would render the results more accessible to a rather general reader, like myself.

We thank the reviewer for their positive comments and for recommending our work for publication in Nature Communications.

Further questions/comments

1. Could the electric field used to make skyrmions mobile? I think this would certainly increase the impact of the article.

We thank the referee for this comment. While we agree that probing skyrmion mobility would be a valuable addition to our work, our primary goal of our study is to demonstrate that non-chiral banana-shaped particles governed purely by hard-core interactions can assemble into blue phases and skyrmions due to the bend flexoelectric effect. As stated in our article, this study constitutes the first evidence of this phenomenon through particle-based simulations. It not only validates previous theoretical conjectures but it also highlights how the emergence of a double twist from a simple particle shape leads to intricate modulated phases beyond the twist-bend nematic phase. We therefore consider it to be outside the scope of this paper to devise protocols that makes these skyrmions mobile. Furthermore, our particle-based simulations will be computationally too expensive to probe the skyrmion dynamics under an oscillatory electric field over long time scales. We hope that we can address this question in future work by extending our Landau-de Gennes approach that we developed recently for lyotropic banana-shaped particles [The Journal of chemical Physics 152, 224502 (2020)]. We hope our findings will inspire future research aimed at exploring their potential applications in novel technologies.

We mention that this will be an interesting question to address in future work:

We hope that our findings will inspire experimental investigations of bend-stabilized 3D BPIII and 2D skyrmion phases in both thermotropic and lyotropic liquid crystals. Furthermore, it will be interesting to explore ways to manipulate these skyrmions as well as their mobility for potential applications in future work.

2. I do not quite understand the isolated skyrmion phase. Is it really just one, or would this depend on the size of the system? In which case it might be some kind of isotropic skyrmion state?

We agree with the reviewer that the term "isolated skyrmions" is misleading. As the referee pointed out, this phase should be regarded as an isotropic, gas-like phase of skyrmions, as initially discovered by Nych *et al.* [7] in their study of chiral liquid crystals under confinement. The term does not imply a single skyrmion. Although "skyrmion gas" might better describe this phase, we prefer to maintain consistency with the commonly used terminology. We have clarified this term more extensively in the revised manuscript to avoid any potential misinterpretation.

We modified the description of the IS phase to emphasize its isotropic nature.

As the thickness increases to $L_{ee} < H < 2r_s$, with r_s the radius of the skyrmion filaments, the banana-shaped particles nucleate DTCs that align perpendicularly to the walls. At sufficiently low confinement heights, only a few isolated skyrmions (IS) are randomly nucleated, forming a dilute, gas-like phase of skyrmions.

3. I am bit confused with the achirality of the observed states. I guess one could have either, a collapse of chirality such that all the twists within the system have the same handedness, while on averaging over all the samples, a racemic results is recovered. Another, way, which I think is the case here, is to have a racemic state where half the twists are right and other half are left handed. However, looking at the figure S1 in the supplement, shows non-racemic handedness of the skyrmion filaments.

We thank the referee for this nice question, which allows us to clarify an important point. The skyrmion lattice in our system is indeed a racemic mixture of left-handed and right-handed skyrmions. Figure S1 shows this mixture, with a slight asymmetry in the abundance of one handedness over the other, due to fluctuations arising from the small system size. This emphasizes the irrelevance of skyrmion chirality in their interactions, representing the underlying physics. The formation of a hexagonal lattice of skyrmions SL is a direct consequence of the achirality of the effective interaction between skyrmions: regardless of chirality, particles on the edge of a half-skyrmion align their long axes perpendicular to the skyrmion axis, and point their polar axes inward. This makes the skyrmion-skyrmion interaction independent of chirality, allowing the formation of a hexagonal lattice. In contrast, in the case of quarter-skyrmions (see Ref. [8]), where particles at the boundary lie at 45° with respect to the axis, chirality becomes crucial in regulating skyrmion interactions. Only skyrmions with opposite chirality can neighbour each other. The resulting equilibrium lattice would then be an "antiferromagnetic" square lattice with a perfect balance between left- and right-handed skyrmions.

We have clarified this more precisely in the revised manuscript:

As the thickness increases to $L_{ee} < H < 2r_s$, with r_s the radius of the skyrmion filaments, the banana-shaped particles nucleate DTCs that align perpendicularly to the walls. At sufficiently low confinement heights, only a few isolated skyrmions

(IS) are randomly nucleated, forming a dilute, gas-like phase of skyrmions. The mobility of the vortices is determined by the available space to relax the DTC pitches, and at larger thicknesses, the DTCs pack into a quasi-2D lattice of hexagonally packed skyrmions (SL). It is important to note that the skyrmion lattice in our system is a racemic mixture of left-handed and right-handed skyrmions. Figure S1 shows this mixture, with a slight asymmetry in handedness due to small system size fluctuations. The formation of a hexagonal skyrmion lattice (SL) results from the achirality of the skyrmion-skyrmion interaction: regardless of chirality, particles on the edge of a half-skyrmion align their long axes perpendicular to the skyrmion axis and point their polar axes inward. This makes the skyrmion interaction independent of chirality, allowing the formation of a hexagonal lattice. In contrast, for quarter-skyrmions (see Ref. [8]), where particles at the boundary lie at 45° with respect to the axis, chirality becomes crucial in skyrmion interactions, allowing only skyrmions with opposite chirality to neighbour each other. The resulting equilibrium lattice would then be an "antiferromagnetic" square lattice with a perfect balance between left- and right-handed skyrmions. Finally, as the thickness approaches the equilibrium pitch, another phase forms, where long and flexible DTCs are bounded by the confining walls. As it will be discussed next, such a phase is reminiscent of the BP111, which to date, has only been found in thermotropic cholesteric liquid crystals.

-
- [1] Chiappini, M.; Dijkstra, M., *Nat. Commun.*, **2021**, *12*, 2157. doi: 10.1038/s41467-021-22413-8
- [2] Pišljarić, J.; Ghosh, S.; Turlapati, S.; Rao, N.; Skarabot M.; Mertelj, A.; Petelin, A.; Nych, A.; Marinčič, A.; Pusovnik, A.; et al., *Phys. Rev. X*, **2022**, *12*, 011003.
- [3] Shamid, S. M.; Allender, D. W.; Selinger, J. V., *Phys. Rev. Lett.*, **2014**, *113*, 237801. doi: 10.1103/PhysRevLett.113.237801
- [4] Alexander, G.; Yeomans, J., *Phys. Rev. Lett.*, **2007**, *99*, 067801. doi: 10.1103/PhysRevLett.99.067801
- [5] Shamid, S. M.; Dhakal, S.; Selinger, J. V., *Phys. Rev. E*, **2013**, *87*, 052503. doi: 10.1103/PhysRevE.87.052503
- [6] Kubala, P.; Tomczyk, W.; Ciésła, M., *J. Mol. Liq.*, **2022**, *367*, 120156. doi: 10.1016/j.molliq.2022.120156
- [7] Nych, A.; Fukuda, J.-i.; Ognysta, U.; Žumer, S., *Nat. Phys.*, **2017**, *13*, 1215. doi: 10.1038/nphys4245
- [8] Duzgun, A.; Selinger, J. V.; Saxena, A., *Phys. Rev. E*, **2018**, *97*, 062706. doi: 10.1103/PhysRevE.97.062706
- [9] de Gennes, P. G., *The physics of liquid crystals*, Clarendon Press Oxford, 1974. pp. xi, 333 p.

REVIEWERS' COMMENTS

Reviewer #1 (Remarks to the Author):

The authors have addressed satisfactorily all of comments and I am happy to recommend the publication of this work to Nature Communications.

Reviewer #3 (Remarks to the Author):

I thank the authors for their detailed and pedagogical responses to my questions. I am happy to recommend publication in the current format.